# Efficient Prompting via Dynamic In-Context Learning

## Abstract

In context learning has become a common practice for prompting generalist models. Despite being effective, in-context learning can be computationally inefficient because it makes the input prompt much longer, consuming valuable space in the context window and leading to larger computational costs. In this paper, we propose **DynaICL**, a recipe for efficient prompting with *black-box* generalist models that dynamically allocates in-context examples according to the input complexity and the computational budget. We train a meta controller that predicts the number of in-context examples suitable for the generalist model to make a good prediction based on the difficulty of a specific input. We then dynamically allocate the number of demonstrations for an input according to the computation budget. Experimental results show that DYNAICL helps achieve a better performance-efficiency trade-off in two practical settings where we have constraints on computational resources or the minimum required performance. Specifically, DYNAICL saves up to 46% token budget compared to the common practice that allocates the same number of in-context examples to each input. In addition, we also find that a meta controller trained on a certain backbone model and tasks can successfully generalize to unseen models and tasks, suggesting that we can train a meta controller once and use it in various use cases.

## 1 Introduction

The field of artificial intelligence and natural language processing is witnessing a major paradigm shift from training and deploying multiple specialist models for specific tasks to pre-training one generalist model (e.g., a large language model (LLM)) and prompting for different tasks (Radford et al., 2018; 2019; Brown et al., 2020; Chowdhery et al., 2022; Ouyang et al., 2022; OpenAI, 2023; Zhang et al., 2022; Touvron et al., 2023). While prompting is an elegant and effective way to utilize generalist models, the computational cost remains a major bottleneck. We identify two key sources of the computational inefficiency of prompting generalist models: ***model size*** and ***sample size***. The former is arguably a prerequisite for generalist models to solve all kinds of tasks via prompting and there already exist a number of model compression techniques (Sanh et al., 2020; Michel et al., 2019; Dettmers et al., 2022; Xu et al., 2020) that aim to reduce the size of generalist models. One obvious limitation of these approaches is that they all require the user to train or deploy their own models, and most of them assume the users have access to the model parameters.

In this paper, we instead focus on reducing ***sample size***, a relatively new perspective for improving the efficiency of ***black-box*** generalist models of which the parameters are unavailable to users. This particular direction has received relatively limited exploration within the era of specialist models, as the inputs and outputs associated with it are clearly defined and largely devoid of redundancy. This is no longer true in the context of prompting generalist models such as LLMs because we have a lot of different ways to prompt a model that results in prompts of different lengths. We identify the main factor influencing the prompt length to be *the use of in-context learning* and *the number of in-context examples (demonstrations) in the prompt*. Specifically, in-context learning (Brown et al., 2020) refers to the practice of adding a few exemplar input-output pairs that are related to the input, which helps the generalist model better understand and solve the problem. Although it is still unclear how in-context examples help a generalist model (Min et al., 2022; Yoo et al., 2022; Dai et al., 2022), it is evident that samples of greater complexity necessitate a greater number of in-context examples for a generalist model to acquire contextual understanding. Conversely, simpler

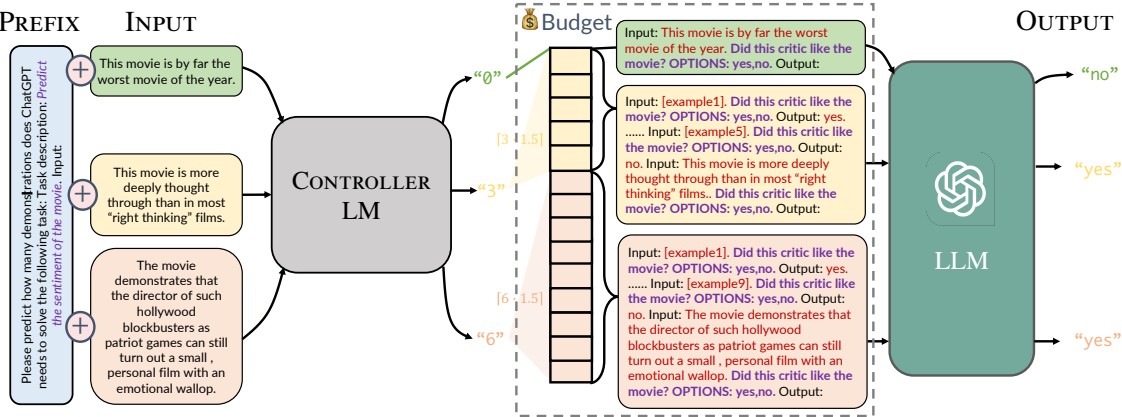

Figure 1: Overview of the DYNAICL framework. Given a set of samples and a token/computation budget, a meta controller first predict a number of in-context examples suitable for each sample. The predictions are then normalized and adjusted according to the budget. We then append the corresponding number of in-context examples to the original prompt. The prompts are then fed into a generalist model to generate predictions.

samples may be solvable even without relying on in-context learning. This is confirmed by our preliminary study, which also finds that assigning more in-context examples to simple samples occasionally confuses the generalist model and turns its prediction from correct to erroneous. These findings suggest that the current practice of allocating a fixed number of in-context examples for all inputs is sub-optimal.

To this end, we propose **Dyna**mic **I**n-**C**ontext **L**earning (**DynaICL**), a dynamic computation framework for prompting generalist models. DYNAICL is conceptually similar to previous work on input adaptive computation for specialist models (Han et al., 2021; Graves, 2017; Teerapittayanon et al., 2016; Schwartz et al., 2020b; Zhou et al., 2020; Huang et al., 2023). The main difference is that DYNAICL aims to dynamically adjust the size of the input while previous work focuses on adjusting the complexity of the model. This results in a major advantage of DYNAICL: it only operates on inputs, thus is disentangled with model architectures or parameters, and suits an increasingly common scenario in the era of generalist models where the users do not have access to the model's parameters. To achieve this, we train a meta controller that predicts the number of in-context examples suitable for the generalist model to make a good performance-efficiency trade-off given a specific input. The meta controller can be instantiated with a smaller pre-trained model (e.g., FLAN-T5 (Wei et al., 2022)) and multi-task fine-tuned with the combination of supervised learning with a novel data synthesis algorithm and reinforcement learning with rewards based on performance-efficiency trade-off. Then at test time, we can dynamically allocate the number of demonstrations for an input according to both the predictions from the meta controller and the computation budget. We illustrate the procedure of efficient prompting with DYNAICL in Figure 1.

We test the effectiveness of DYNAICL in the context of prompting LLMs due to its prominence as the predominant use case for generalist models at present. We experiment with ChatGPT as the generalist model and train a meta controller on a subset of the FLAN dataset collection (Longpre et al., 2023). We evaluate DYNAICL in two practical settings where either the computational resources or the performance is under constraints. We find that compared with the common practice of uniformly allocating in-context examples, DYNAICL can achieve an averaged absolute performance improvement of 2.6% within a certain computational budget or reach a certain performance requirement with up to 46% less compute (in terms of total token consumption) across 8 tasks. We also find that a meta controller trained on certain tasks with a certain generalist model (i.e., ChatGPT) can generalize well to unseen tasks (even with different output formats) and other generalist models (e.g., LLAMA (Touvron et al., 2023)). To the best of our knowledge, our work is among the first approaches that can accelerate a black-box generalist model without access to its parameters.

## 2 Methodology

### 2.1 Background: In-Context Learning

We first recall some basics of prompting and in-context learning. Prompting refers to the process of providing a prompt, which typically contains a description of the task and the task input, to a generalist model that guides its response generation. Formally, let $\mathcal{G}$ be a generalist model and $P$ be a prompt. Then, the output $O$ is given by: $O = \mathcal{G}(P)$. Prompting relies on the generalist model's ability to understand and follow abstract instructions, which sometimes leads to unsatisfactory empirical performance, especially for hard tasks that require complex reasoning.

On the other hand, in-context learning leverages the ability of a generalist model to adapt to new information provided within the input context. Formally, given N labeled examples $\{(x_i, y_i)\}_{i=1}^{N}$ and a hand-crafted template $\mathcal{T}$, in-context learning first verbalizes each input-output pair with a template, resulting in demonstrations $d_i = \mathcal{T}(x_i, y_i)$. Then the generalist model takes the concatenation of the original prompt and the demonstrations to generate the output:

$$O = \mathcal{G}(P \oplus d_1 \oplus d_2 \oplus \cdots \oplus d_N) \tag{1}$$

where $\oplus$ denotes the concatenation of token sequences.

### 2.2 Meta Controller

**Architecture and Input/Output Formats:** The meta controller $\mathcal{C}$ can be modeled by any sequence generation model including both encoder-decoder models and decoder-only models. We use an instruction-tuned model such as FLAN-T5 as the backbone for the meta controller to facilitate training. As illustrated in Figure 1, it receives a task instruction and an input, which is identical to most instruction tuning literature (Sanh et al., 2022; Wei et al., 2022; Taori et al., 2023). But instead of generating the corresponding outputs like instruction-tuned models, our meta controller is trained to generate the number of in-context examples suitable for the input to achieve the best performance-efficiency trade-off, which we denote as $k$. This process can be expressed by $k = \mathcal{C}(P)$. The output expresses the confidence modeling of the meta controller for the generalist model to some extent. This method pertains to, albeit distinguishes itself from, prior existing work on model calibration (Guo et al., 2017; Kadavath et al., 2022), which addresses the inherent confidence levels of the model itself.

**Training** We then present our two-stage training framework for the meta controller. In the first stage, we train the meta controller to predict the minimum number of in-context examples for the generalist model to produce a good output. "A good output" can have different definitions for different tasks. For example, it can be defined as predicting the correct label with a high probability for classification tasks and generating outputs similar to the ground truth for generation tasks. In this paper, we consider only classification tasks following (Hao et al., 2022; Li et al., 2023b). To synthesize training data for supervised training, we propose a simple and intuitive data generation method. Specifically, for a prompt $P$, we consider the minimum number of in-context examples $k^*$ for it to be the number that makes the generalist model's expected accuracy exceed a certain (hand-crafted) threshold $t$:

$$k^* = \min_{k \in \mathbb{N}} \left\{ k \mid \mathbb{E}_{(x_i, y_i)^k \sim \mathcal{D}^k} \left[ \mathrm{Acc}(\mathcal{G}(P, \mathcal{T}(x_{1:k}, y_{1:k}))) \right] > t \right\} \tag{2}$$

where $\mathcal{D}^k$ denotes all subsets of the training data of size $k$ and $\mathrm{Acc}(\mathcal{G}(P, \mathcal{T}(x_{1:k}, y_{1:k})))$ denotes the performance (e.g., accuracy) of model $\mathcal{G}$ using template $P$ and in-context examples $(x_1, y_1) \cdots (x_k, y_k)$.

We synthesize $(P, k^*)$ pairs on a mixture of instruction-tuning datasets from the FLAN collection and train the meta controller with maximum likelihood estimation.

After the first stage, the meta controller can already predict a reasonable number of in-context examples for a prompt. However, we may want it to better satisfy a certain performance-efficiency trade-off in a more fine-grained way. To this end, we propose to fine-tune the meta controller with reinforcement learning

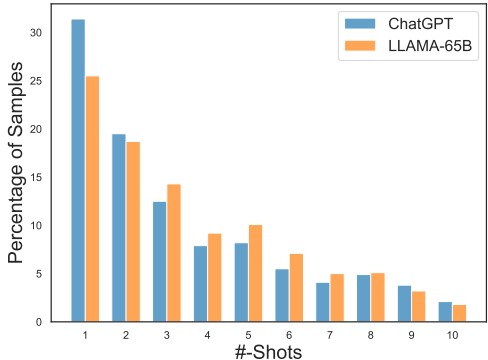

| $\Delta$ **Accuracy** | ✗$\to$ ✓ | ✓$\to$ ✗ |
|---|---|---|
| ***zero-shot $\to$ 1-shot*** | | |
| + 2.5% | 3.9% | 1.4% |
| ***1-shot $\to$ 5-shots*** | | |
| + 1.4% | 1.9% | 0.5% |
| ***5-shots $\to$ 64-shots*** | | |
| + 0.3% | 0.7% | 0.4% |

Figure 2: Distribution of the number of in-context examples that suffice for making the correct prediction for samples that cannot be answered correctly by zero-shot inference with generalist models but can be solved with in-context learning for up to 10 shots. The generalist model we consider are ChatGPT and LLAMA-65B, and the dataset is CSQA.

Figure 3: The impact of adding more in-context examples. $\Delta$ **Accuracy** denotes the change of accuracy after adding more in-context examples. ✗$\to$ ✓ and ✓$\to$ ✗ denotes the percentage of examples of which the predictions are changed from incorrect to correct, and vice versa, after adding more in-context examples. We use ChatGPT as the generalist model and TriviaQA as the dataset.

using a reward reflecting the performance-efficiency trade-off. In particular, we define the reward $\mathcal{R}$ to be a linear interpolation of the expected performance (defined as accuracy in case of classification task), and the efficiency, defined as the number of in-context examples $k$:

$$\mathcal{R}(\mathcal{G}, P, k) = \mathbb{E}_{(x_i, y_i)^k \sim \mathcal{D}^k}[\text{Acc}(\mathcal{G}(P, \mathcal{T}(x_{1:k}, y_{1:k}))] + \alpha \cdot k \tag{3}$$

where $\alpha$ is the weight controlling whether the controller should lean towards better performance or efficiency. The meta controller $\mathcal{C}$ is then fine-tuned with policy gradient:

$$\nabla_\theta J(\theta) = \mathbb{E}_{P \sim \mathcal{P}, k \sim \mathcal{C}(k|P,\theta)}[\nabla_\theta \log \mathcal{C}(k|P,\theta)(\mathcal{R}(\mathcal{G}, P, k))] \tag{4}$$

where $\mathcal{P}$ is the set of prompts from a mixture of instruction tuning datasets, and $\mathcal{C}(k|P,\theta)$ denotes the predicted probability mass of $k$ from the meta controller $\mathcal{C}$ for a prompt $P$.

The training framework can be easily adapted for generation tasks by changing the accuracy metric to some generation metrics such as BLEU (Papineni et al., 2002) or BERTScore (Zhang et al., 2020), and doing some normalization to make it compatible with classification tasks. We leave this for future work.

## 2.3 Dynamic In-Context Example Allocation

After training, the meta controller predicts the number of in-context examples for a specific input. This is a naive version of DYNAICL. However, in practice one may have a different computation budget. Therefore it is often desirable to normalize the predictions from the meta controller and dynamically adjust the actual number of in-context examples according to the computation budget. In this work, we propose a simple recipe for dynamic in-context example allocation. Assuming we have a budget of $N$ tokens[1] for $K$ samples. The uniform baseline is to allocate $N/(K \cdot L)$ in-context examples for each sample assuming $L$ is the average length of an example. DYNAICL instead allocates $E$ in-context examples for an input $P$ following:

$$E(P) = [\beta \cdot (\mathcal{C}(P)/\tilde{\mathcal{C}}) \cdot N/(K \cdot L)] \tag{5}$$

where $\mathcal{C}(P)$ is the prediction from the meta controller, $[]$ denotes the rounding operator, $\tilde{\mathcal{C}}$ is the averaged prediction for all examples, and $\beta$ is the token saving ratio ranging from 0 to 1.

---

[1]We consider the budget in terms of the token count because this is the typical scenario for using commercial generalist models such as ChatGPT. We omit the token consumption for the original input for simplicity.

| Models | SST-2 | AGNews | RTE | CB | ARC-E | ARC-C | MRPC | COPA | **Avg. Acc** |
|---|---|---|---|---|---|---|---|---|---|
| *zero-shot* | | | | | | | | | |
| ChatGPT | 88.5 | 84.5 | 84.5 | 89.5 | 85.1 | 61.0 | 88.4 | 67.2 | 81.1 |
| *Budget: 5-shots on average* | | | | | | | | | |
| Uniform | 93.2 | 87.9 | 86.1 | 91.1 | 88.3 | 64.8 | 90.4 | 88.2 | 86.2 |
| Random | 93.0 | 87.7 | 86.1 | 91.0 | 88.1 | 65.0 | 90.4 | 89.4 | 86.3 |
| **DynaICL** | **95.3** | **90.2** | **88.1** | **92.9** | **90.5** | **68.4** | **91.8** | **93.0** | **88.8** |
| *Budget: 10-shots on average* | | | | | | | | | |
| Uniform | 95.8 | 90.9 | 88.5 | 93.1 | 90.8 | 68.3 | 92.0 | 93.4 | 89.1 |
| Random | 95.9 | 90.7 | 88.4 | 93.3 | 90.8 | 68.2 | 92.1 | 92.8 | 88.9 |
| **DynaICL** | **96.7** | **92.5** | **90.0** | **94.1** | **91.9** | **70.0** | **93.1** | **95.8** | **90.5** |

Table 1: Main results on *seen* tasks during meta controller training. The total computation/token budget is the same inside each group. DYNAICL consistently outperforms all baselines across all tasks and budgets.

## 3 Experiments

In this section, we test the empirical effectiveness of DYNAICL by experimenting on some NLP tasks with ChatGPT, a popular large language model, as the generalist model. We first describe the experimental settings. Then we begin with a preliminary study about the impact of the number of in-context examples to motivate our approach. After that, we evaluate DYNAICL by answering two research questions for two realistic settings:

- **RQ1:** To what extent can DYNAICL improves the performance of a generalist model with fixed computational budgets?

- **RQ2:** To what extent can DYNAICL reduce computational cost or token consumption for a generalist model to achieve a fixed target performance?

### 3.1 Experimental Settings

**Models** We consider ChatGPT as the generalist model for training the meta controller and the main experiments. We use LLAMA-65B as an unseen generalist model for evaluating the generalization ability of the meta controller. We use FLAN-T5-large, which has less than 1B parameters, to initialize the meta controller. We also test with FLAN-T5-base in the analysis.

**Tasks** We use a subset in the FLAN collection containing 30+ classification tasks to train the meta controller. For evaluation, we test DYNAICL on both *seen* and *unseen* tasks, which are explicitly excluded from the training data for the meta controller. To be specific, we use SST-2 (Socher et al., 2013), AGNews (Zhang et al., 2015), RTE (Dagan et al., 2006; Haim et al., 2006; Giampiccolo et al., 2007; Bentivogli et al., 2009), CB (De Marneffe et al., 2019), ARC-E (Clark et al., 2018), ARC-C (Clark et al., 2018), MRPC (Dolan & Brockett, 2005), and COPA (Roemmele et al., 2011) as the seen tasks, and PIQA (Bisk et al., 2020), OpenBookQA (Mihaylov et al., 2018), CommonsenseQA (Talmor et al., 2019), TriviaQA (Joshi et al., 2017), Natural Questions (Kwiatkowski et al., 2019), and Web Questions (Berant et al., 2013) as unseen tasks. It is noteworthy that TriviaQA, Natural Questions, and Web Questions are not classification tasks but a trained meta controller can still be used despite being trained only on classification tasks. This is because its input format (i.e., instruction + input) is agnostic to the type of the task.

**Training Details** We follow Wei et al. (2022) and fine-tune the meta controller for 30k/5k gradient steps with a batch size of 8,192 tokens using the Adafactor Optimizer (Shazeer & Stern, 2018) with a learning rate of 3e-5/1e-5, for the first/second training stage, respectively.

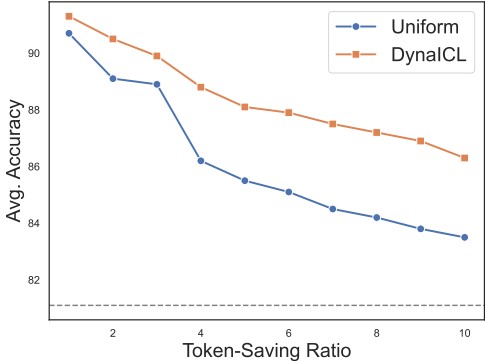

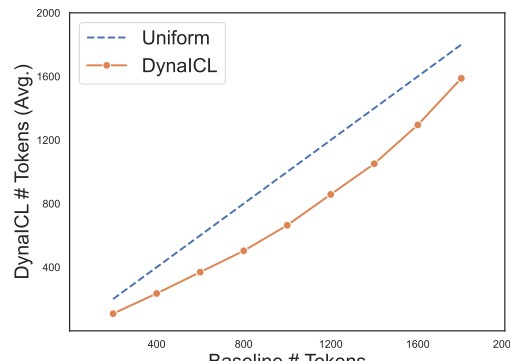

(a) Performance comparison between DYNAICL and the uniform baseline under different token saving ratios defined as the ratio between actual token usage and the token usage of using 20 in-context examples per sample. The accuracy is averaged across all seen test datasets. The dashed line is the zero-shot performance.

(b) Token saving ratio of DYNAICL compared to the uniform baseline under performance constraints defined by the performance of the uniform baseline with different token budgets. Each point (x,y) in the line indicates that on average, DYNAICL needs to use $y$ tokens to match the performance of the uniform baseline with $x$ tokens.

Figure 4: Performance comparison when fixing either the compute budget or the target performance.

**Baselines**  We mainly compare DYNAICL with the uniform baseline that allocates the same number of in-context examples for each sample, and the random baseline that randomly samples a number of in-context examples from a Gaussian distribution. We only compare these two naive baselines because there is no prior work in this direction and popular methods for efficient NLP can not be applied in this setting.

## 3.2 Preliminary Study: How Much Do More In-Context Examples Help?

We first conduct a preliminary study investigating the role of adding more in-context examples to the prompt for different samples. We first test if most samples for a task require a similar amount of in-context examples for a generalist model to generate a good output. We plot the distribution of the number of in-context examples that suffice for making the correct prediction for samples from the CommonsenseQA dataset that cannot be answered correctly by zero-shot inference with ChatGPT or LLAMA-65B but can be solved with in-context learning for up to 10 shots. As shown in Figure 2, different samples requires a very different amount of in-context examples. Some hard examples require 10 in-context examples for a generalist model to make the correct prediction while most examples require only one in-context example or can be solved with zero-shot inference. This observation confirms the necessity of dynamically allocating in-context examples according to sample difficulties. Moreover, we can see that ChatGPT and LLAMA-65B share similar trends in the Figure. This suggests that a meta controller trained with one generalist model may be able to generalize to other generalist models, which is later proved in our analysis.

Then we further analyze the effect of scaling more in-context examples. As shown in Figure 3, the effectiveness of adding more in-context examples to the prompt is amortized when there are already a few (e.g., 5) in-context examples. This also supports our motivation that only a few samples require many in-context examples and uniformly allocating an equal number of in-context examples for all samples is a waste of tokens and computation. More interestingly, we find that sometimes it can be harmful to include more in-context examples for a sample that can already be correctly solved by the generalist model, which is shown by a non-negligible amount of samples' predictions are changed from correct to incorrect after adding more in-context examples. This further confirms the potential of DYNAICL to achieve better performance while consuming fewer tokens.

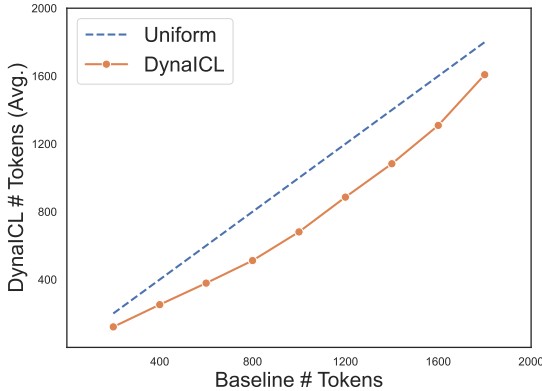
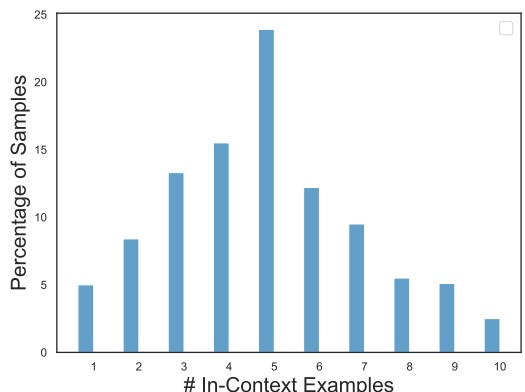

(a) Token saving ratio of DYNAICL compared to the uniform baseline under different performance constraints on seen tasks. DYNAICL is trained with ChatGPT but tested with LLAMA-65B.

(b) Distribution of samples (on seen tasks) according to the number of in-context examples allocated for them. The computational budget is fixed to 5 in-context examples per sample.

Figure 5: Analysis on the generalization ability of DYNAICL on unseen generalist models and the distribution of samples according to the number of in-context examples allocated for them.

### 3.3 Main Results

We first compare the performance of DYNAICL with the baselines in Table 1. We can see that DYNAICL leads to an averaged performance improvement of 2.6% and 1.4% over the uniform baseline with budgets of 5 and 10 in-context examples for each sample, respectively. This confirms that DYNAICL leads to improved performance with fixed budgets. We also plot the trend of averaged performance on seen tasks with different token-saving ratios in Figure 4 (a). We can see that DYNAICL leads to consistent improvements across all budgets and the improvements are larger when the computation/token budget is more limited. We then show the extent to which DYNAICL can save tokens for achieving a fixed target performance in Figure 4 (b). We can see that DYNAICL consistently require fewer tokens to match the performance achieved by the uniform baseline with certain budgets. Specifically, DYNAICL only consumes 108 tokens on average to match the performance of the common practice with 200 tokens on average. This confirms that DYNAICL can effectively reduce token/computation consumption for achieving a fixed target performance.

### 3.4 Analysis

We then conduct an analysis investigating the impact of different components in DYNAICL and the generalization ability of DYNAICL on unseen tasks or generalist models when training the meta controller.

**Ablation Study** We first analyze the impact of the two training stages, the size of the meta controller, and the number of tasks the meta controller is trained with. The results are shown in Table 2. We find that both training stages contributes to the performance of DYNAICL and the first stage is more important. We think this is because the first training stage provides an important starting point for the second stage using reinforcement learning. We also find that DYNAICL with a smaller meta controller or a meta controller train on fewer tasks also achieves competitive performances.

**Generalization on Unseen Tasks** We then test how well DYNAICL can generalize on unseen tasks. The results are shown in Table 3. We find that DYNAICL consistently leads to performance improvements across all 6 unseen tasks. Notably, DYNAICL also leads to substantial improvements on Natural Questions and Web Questions, which are generative question answering datasets that are very different from text classification tasks during training. This confirms that DYNAICL can generalize well on tasks that are not used to train the meta controller.

| Models | SST-2 | AGNews | RTE | CB | ARC-E | ARC-C | MRPC | COPA | **Avg. Acc** |
|---|---|---|---|---|---|---|---|---|---|
| *Budget: 5-shots on average* | | | | | | | | | |
| Uniform | 93.2 | 87.9 | 86.1 | 91.1 | 88.3 | 64.8 | 90.4 | 88.0 | 86.2 |
| **DynaICL** | **95.3** | **90.2** | **88.1** | **92.9** | **90.5** | **68.4** | **91.8** | **93.0** | **88.8** |
| - first stage | 93.8 | 88.4 | 86.6 | 91.8 | 89.1 | 65.5 | 90.8 | 89.6 | 86.9 |
| - second stage | 94.4 | 89.5 | 87.5 | 92.1 | 89.5 | 67.1 | 91.2 | 91.4 | 87.8 |
| w/ smaller model | 94.8 | 89.2 | 87.5 | 92.3 | 90.2 | 67.7 | 91.3 | 92.2 | 88.2 |
| w/ fewer tasks | 95.0 | 89.3 | 87.3 | 92.5 | 90.0 | 68.0 | 91.5 | 92.4 | 88.3 |

Table 2: Ablation study results. "- first stage" and "- second stage" denotes the ablated variants where the meta controller is not trained with the first or second stage training, respectively. "w/ smaller model" and "w/ fewer tasks" denotes the ablated variants where the meta controller is parameterized with FLAN-T5-Base and the meta controller is trained with 50% less training tasks.

| Models | PIQA | OBQA | CSQA | TriviaQA (EM) | NaturalQ (EM) | WebQS (EM) | **Avg.** |
|---|---|---|---|---|---|---|---|
| *zero-shot* | | | | | | | |
| ChatGPT | 83.3 | 60.9 | 74.5 | 80.2 | 27.5 | 22.9 | 58.2 |
| *Budget: 5-shots on average* | | | | | | | |
| Uniform | 84.3 | 61.5 | 76.6 | 84.1 | 37.1 | 26.3 | 61.6 |
| **DynaICL** | **85.4** | **62.8** | **77.2** | **84.4** | **40.2** | **28.8** | **63.1** |
| *Budget: 10-shots on average* | | | | | | | |
| Uniform | 85.9 | 63.1 | 77.4 | 84.3 | 40.8 | 29.2 | 63.4 |
| **DynaICL** | **86.3** | **63.7** | **77.9** | **84.5** | **42.4** | **29.9** | **64.1** |

Table 3: Analysis of the generalization ability of DYNAICL on datasets that are *unseen* when training the meta controller. Tasks with (EM) suffix denotes the task is generative question answering and we use exact match as the metric. DYNAICL still consistently outperforms the baseline across all tasks.

**Generalization on Unseen Generalist Models** We also test if DYNAICL can generalize to other generalist models that are not used for training the meta controller by applying the meta controller trained with ChatGPT with LLAMA-65B as the generalist model. Results in Figure 5 (a) show that DYNAICL still saves a great number of tokens for achieving the same performance with the uniform baseline even tested with a different generalist model. This confirms that DYNAICL can generalize well on generalist models that are not used to train the meta controller.

**Distribution of In-context Examples Count** We then plot the distribution of samples according to the number of in-context examples allocated for them to better understand the meta controller. As shown in Figure 5 (b), with a target budget of 5 in-context examples, a large portion of samples are allocated with 5 in-context examples in DYNAICL. This indicates that most samples are predicted to need a similar number of in-context examples as the averaged prediction. We also find that more samples are assigned with fewer than 5 in-context examples while a few hard samples are assigned with more in-context examples. We present a qualitative study of different samples and the corresponding number of in-context examples allocated to them in the Appendix.

**Computation Cost of the Meta Controller** Finally, it is noteworthy that the meta controller does add some computational cost and latency overhead to the overall prompting procedure. However, since the meta controller can use a very small backbone such as T5-large or T5-base, its computation cost is negligible compared to that of a generalist model. To be specific, the computational cost (in terms of FLOPs) of a T5-large based meta controller for a sample of 50 tokens is less than 0.1% of the change of the computation

cost when changing the input from 200 tokens to 199 tokens, or less than 0.0005% of the computational cost saved by reducing one in-context example from the prompt. Similarly, since the meta controller only needs to predict 1 or 2 tokens, the latency overhead accounts for only 0.1% to 0.2% of the latency of calling the GPT-3.5-turbo API, and reducing one in-context example will lead to a speedup of around 10%. In sum, we believe the computational and latency overhead from the meta controller is almost negligible.

## 4 Related Works

Training a generalist model that can solve a wide range of tasks without task-specific training has been a long-standing goal in the field of artificial intelligence. One pioneering work dates back to Collobert & Weston (2008) that attempted to solve all NLP tasks with a shared architecture using multi-task learning. This idea is further improved by decaNLP (McCann et al., 2018) that proposes to convert all NLP tasks to question answering format. T5 (Raffel et al., 2020) then improves this paradigm by using text-to-text format for unifying all NLP tasks, which is more general and friendly to scaling. Finally, GPT-3 (Brown et al., 2020) show that by scaling model size, training data, and training FLOPs, a large language model can serve as a generalist model that solves many tasks by simply writing a prompt that describes the task and the input. They also showed that the zero-shot ability of a large language model can be further improved by adding a few input-output demonstrations in the prompt to help the model better understand the task. Since then, a large number of work has been done for improving and understanding prompting and in-context learning with large language models. For instance, Schick & Schütze (2021) show that small encoder models can also be prompted. Min et al. (2022) show that in-context examples mainly help a generalist model learn output label space and distribution of input text. Kadavath et al. (2022) prove that generalist models are well calibrated and can be trained to model their confidence level. Hao et al. (2022) and Li et al. (2023b) show that in-context learning with many examples improves the overall performance of a generalist model.

### 4.1 Generalist Models, Prompting, and In-context Learning

Training a generalist model that can solve a wide range of tasks without task-specific training has been a long-standing goal in the field of artificial intelligence. One pioneering work dates back to Collobert & Weston (2008) that attempted to solve all NLP tasks with a shared architecture using multi-task learning. This idea is further improved by decaNLP (McCann et al., 2018) that proposes to convert all NLP tasks to question answering format. T5 (Raffel et al., 2020) then improves this paradigm by using text-to-text format for unifying all NLP tasks, which is more general and friendly to scaling. Finally, GPT-3 (Brown et al., 2020) show that by scaling model size, training data, and training FLOPs, a large language model can serve as a generalist model that solves many tasks by simply writing a prompt that describes the task and the input. They also showed that the zero-shot ability of a large language model can be further improved by adding a few input-output demonstrations in the prompt to help the model better understand the task. Since then, a large number of work has been done for improving and understanding prompting and in-context learning with large language models. For instance, Schick & Schütze (2021) show that small encoder models can also be prompted. Min et al. (2022) show that in-context examples mainly help a generalist model learn output label space and distribution of input text. Kadavath et al. (2022) prove that generalist models are well calibrated and can be trained to model their confidence level. Hao et al. (2022) and Li et al. (2023b) show that in-context learning with many examples improves the overall performance of a generalist model. Recently, the paradigm of prompting generalist models is successfully transferred to other modalities other than language. For example, Zhou et al. (2022) and Li et al. (2023a) explored prompting vision models. It is foreseeable that prompting generalist models will become the de-facto paradigm for most domains in artificial intelligence.

### 4.2 Efficient Deep Learning

Improving the performance-efficiency trade-off of deep learning models is a very interesting research problem and is attracting more and more attention since the rise of large generalist models (Xu et al., 2021; Schwartz et al., 2020a). A large number of work has been done on improving the speed and efficiency of large language models, including both *static* methods such as knowledge distillation (Hinton et al., 2015; Romero et al.,

2015; Sanh et al., 2020), pruning (LeCun et al., 1989; Michel et al., 2019), quantization (Han et al., 2016; Shen et al., 2020; Dettmers et al., 2022) and module replacing (Xu et al., 2020); and *dynamic* methods such as adaptive computation (Graves, 2017), early-exiting (Teerapittayanon et al., 2016; Schwartz et al., 2020b; Zhou et al., 2020), and model cascade (Li et al., 2021; Varshney & Baral, 2022).

However, most of the aforementioned methods require access to the model parameters, which may not be possible for closed source models. One exemption is model cascade, which first sends the input to a cheaper model and optionally sends it to a more powerful model if the previous model is not confident enough. This method, however, also faces server latency issues because harder samples will be computed by multiple generalist models sequentially. Concurrently to our work, Mu et al. (2023) proposes to train gist tokens to replace long-form prompts and show promising results on prompt compression. However, this approach is still limited to white-box settings where the model parameter is available and also compromises interpretability.

## 5 Conclusions

This paper introduces DYNAICL, a framework for efficiently prompting generalist models. We propose to train a meta controller that predicts the suitable number of in-context examples for a specific sample with a two-stage training framework. During inference, DYNAICL dynamically allocate different number of in-context examples to samples according to the predicted difficulty and the computational budget. Our experiments show that DYNAICL consistently leads to better performance-efficiency trade-offs across tasks, models, and scenarios. We also find a meta controller trained on a collection of around ten tasks can successfully generalize to tasks unseen during training.

## 6 Ethics Statement

As for technical limitations, the main limitation of this work is that we only test DYNAICL on NLP tasks with LLMs as the backbone, while it may also be interesting to test on other modalities such as vision tasks with multi-modal generalist models. This is because the main experiments are conducted before multi-modal instruction following models such as LLAVA came out. We leave this for future work. Another limitation is that we only train the meta controller with text classification datasets. We explain how the meta controller can be trained on generation tasks at the end of Section 2.2. We also experiment with some generative question answering datasets and show DYNAICL trained only on classification tasks can successfully transfer to these tasks. Finally, the dynamic in-context example allocation algorithm is quite naive. Potential improvements may be made using some more sophisticated planning or optimization algorithms. We also leave this for future work.

As for social impact, this work aims to reduce the token/computation consumption of prompting generalist models. It probably leads to a positive environmental impact and will unlikely lead to any negative social impact.

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

## A   Appendix

We present a few examples of how many in-context examples DynaICL allocates to different samples in the SST-2 dataset with an average budget of 5 in-context examples:

- "it 's disappointing when filmmakers throw a few big-name actors and cameos at a hokey script ." : **1**

- "how did it ever get made ?": **2**

- "not only does the movie fail to make us part of its reality , it fails the most basic relevancy test as well ." : **2**

- "it would n't be my preferred way of spending 100 minutes or $7.00.": **6**

- "but if it is indeed a duty of art to reflect life , than leigh has created a masterful piece of artistry right here .": **7**

    We find that DynaICL does tend to assign fewer in-context examples to easier samples and more in-context examples to harder samples.

