# OpenReview forum: "Efficient Prompting via Dynamic In-Context Learning"
_TMLR — Rejected by TMLR_

### Review · Reviewer_grQg · 2025-06-04

**Summary Of Contributions:**

The paper introduces DynaICL, a method for subsampling examples for in-context learning. The paper proposes to train a surrogate model called a meta controller that dynamically allocates the number of examples according to the input complexity and the computational budget. Results show that DynaICL shows improves performance compared to the baseline, which uses all the in-context examples. Furthermore, they also show that DynaICL reduces the token budget by up to 46% compared to the baseline. Finally, they show that a meta controller trained with one backbone generalizes to new unseen models and tasks.

**Audience:**

Yes

**Claims And Evidence:**

Yes

**Requested Changes:**

See weaknesses.

**Strengths And Weaknesses:**

### Strengths

- The paper is mostly well written. The core idea and motivation come across in the writing.

- The paper introduces a simple data-centric method to train a meta controller that reduces the number of examples in in-context learning.

- DynaICL shows improved performance compared to the baseline. Additionally, the trained meta-controller can also select examples in unseen models which can have a strong practical impact.

### Weaknesses

**DynaICL.**
The paper claims that “reducing sample size” improves in-context learning performance. While I completely agree with this sentence, the paper does not consider the influence of order in in-context examples [d]. This is a confounder in the paper. One issue would be if the performance dropped if the ordering of the examples changed after selecting with DynaICL.

**FLAN dataset.**
The selection of the meta controller training data is unclear. The authors report that they used a subset of the FLAN tasks; however, they do not fully explain how they selected this subset. For example, did they prioritize tasks similar to the target tasks? Furthermore, additional ablation with meta controllers trained on different subsets can be very informative.

**Related work.**
The related work section does not position the paper in relation to other recent work. It would be awesome if you could cite more recent and relevant papers such as [a-c].

**Reproducibility**
The paper presents many of the main results using ChatGPT. While this is acceptable, it raises concerns about reproducibility. It would have been ideal if the authors had tested using a particular version of the GPT API. I acknowledge that the authors have reported results with Llama 65B. Regardless, this is a potential issue with this paper.


References

[a] Many-Shot In-Context Learning, NeurIPS 2024.

[b] Data Curation Alone Can Stabilize In-context Learning, ACL 2023.

[c] Finding Support Examples for In-Context Learning, Findings on EMNLP 2023.

[d] Fantastically Ordered Prompts and Where to Find Them: Overcoming Few-Shot Prompt Order Sensitivity, ACL 2022.

---

### Review · Reviewer_DTfB · 2025-06-25

**Summary Of Contributions:**

This paper proposes a new method called DynaICL, which dynamically selects the number of in-context examples used for ICL to improve efficiency. Specifically, it trains a meta controller that outputs the optimal number of in-context examples per input. Experiments show that, with the average budget fixed, DynaICL achieves better performance compared to baselines with a fixed number of in-context examples.

Strengths
- The problem of improving run time efficiency for ICL is clear and well motivated.
- The core idea of the method is simple and is easy to implement.
- Experiments are convincing, with informative analyses, such as the distribution of samples by the number of in context examples, and evaluation on unseen tasks, which helps understanding the method’s behavior.

Weaknesses
- Experiments are conducted on relatively simple datasets such as classification datasets, where ICL performance saturates quickly and does not see much performance variance across different numbers of in-context examples. The evaluation would be more compelling with more challenging benchmarks or datasets with a larger candidate set, where many-shot ICL is often necessary.
- The claim that the meta controller’s cost is negligible is not entirely convincing. The justification provided is that the controller only outputs one or two tokens. However, increasing the number of ICL examples (e.g., from 5 to 10) also does not increase the output length – only increases the length of the conditioning text. And I do not believe the cost of the meta controller is included in the analysis, e.g., Figure 5(a) (but please correct me if I am wrong). A more systematic analysis of compute cost, such as latency or FLOPs, would be needed.
- While the paper claims that DynaICL generalizes to unseen tasks, the reported gains over the uniform baseline are very marginal (e.g., <1%), so the practical significance of the improvement is unclear.
- Overall, I do not see particular flaws in this paper. However, it’s unclear how impactful this work is in the current context, where ICL is being used less frequently and many models now perform competitively with zero-shot instruction following, which uses far fewer input tokens than ICL.

**Audience:**

Yes

**Claims And Evidence:**

Yes

**Requested Changes:**

- Add experiments where ICL performance has more variance depending on the number of examples, and more challenging benchmarks or datasets with a larger candidate set, where many-shot ICL is often necessary.
- Add efficiency-performance tradeoff results including the meta controller's cost.
- Enhance results on unseen tasks/domains.
- Potentially compare it with instruction following (e.g., prompts that give instructions to perform the task rather than in-context examples)

**Strengths And Weaknesses:**

Written in the previous section

---

### Review · Reviewer_BuE6 · 2025-07-13

**Summary Of Contributions:**

The paper introduces DynaICL, a method to dynamically decide the number of ICL examples during inference. In DynaICL, a “meta-controller” is trained to predict the number of examples required to arrive at the gold answer via a two-step training pipeline consisting of SFT and RL steps. DynaICL has an improved performance-efficiency trade-off than a uniform baseline in 30+ classification tasks.

**Audience:**

Yes

**Broader Impact Concerns:**

no concerns

**Claims And Evidence:**

Yes

**Requested Changes:**

See weaknesses.

It could be helpful to provide more details on how the examples in the prompts are designed.

Additionally, several results in the paper only discuss a single benchmark (e.g., the TriviaQA results in Fig.3), it could be helpful to have all these results in the appendix.

I believe the paper could strongly benefit from -
- Stronger baselines including a comparison to methods that dynamically retrieve examples.
- Experimenting with state-of-the-art models.

**Strengths And Weaknesses:**

### Strengths -

- Reducing the number of tokens in LM prompts is an important problem, and DynaICL leads to improvements in the number of tokens required in ICL.

- The paper is well-written and easy to follow.

- The second-stage of RL training where the reward is defined as an interpolation of the performance and number of ICL examples is elegant and leads to additional improvements (however, the improvements from this stage are relatively small and the effect of the interpolation hyperparameter alpha is not empirically shown).

### Weaknesses -
- My main concern regards the experimental setting and baselines. Regarding baselines, the paper experiments with a uniform sampling baseline and a baseline where the number of examples is predicted at random. However, there are many improved baselines one can think of, for example, sampling more examples for harder tasks (task difficulty can be estimated with an LLM). Regarding the experimental setting, the paper experiments with Llama-65B and ChatGPT (3.5 Turbo, via the API), which are not currently considered state-of-the-art (GPT-3.5-Turbo was released more than two years ago and is considered legacy on the OpenAI API [2]), and are limited to classification tasks, although this is explicitly mentioned as a limitation in the paper.

- Perhaps I missed this, but it was unclear whether the examples in the prompt are chosen at random each time, or shared between all tasks in each dataset. The paper does not discuss or compare to methods that dynamically retrieve examples (e.g., EPR [1]), which can naturally lead to stronger baselines (for example, retrieving only the most relevant examples for each task).

- Empirically, the improvements of DynaICL are relatively small compared to a uniform baseline when generalizing to unseen datasets (+1.5
points 5-shot average, <1 points 10-shot average, Tab.2).

[1] Learning To Retrieve Prompts for In-Context Learning, Rubin et al., NAACL 2022

[2] https://platform.openai.com/docs/models/gpt-3.5-turbo

---

### Decision · Action_Editor_kWyJ · 2025-08-29

**Recommendation:** Reject

**Audience:**

Yes

**Audience Explanation:**

I think the idea of predicting the number of examples has merit and the existing experiments would be interesting to some, but more thorough experimentation is needed.

**Claims And Evidence:**

No

**Claims Explanation:**

The main claim of the paper is that by learning to predict the number of examples one can control the tradeoff of performance vs. compute and reduce the number of tokens without affecting performance considerably. The reviewers point out to various experimental problems in the current manuscript:

a. Baselines: authors do not compare to methods that retrieve relevant examples but uniformly sample.
b. Benchmarks: Classification benchmarks that might not benefit much from high variance in training examples
c. Cost computation: treating the cost of the meta learner as negligible
d. Models: few and relatively outdated models.

**Resubmission Of Major Revision:**

The authors may consider submitting a major revision at a later time.